# Broiler Age Influences the Apparent Metabolizable Energy of Soybean Meal and Canola Meal

**DOI:** 10.3390/ani13020219

**Published:** 2023-01-06

**Authors:** Mahmoud M. Khalil, Mohammad Reza Abdollahi, Faegheh Zaefarian, Peter V. Chrystal, Velmurugu Ravindran

**Affiliations:** 1Monogastric Research Center, School of Agriculture and Environment, Massey University, Palmerston North 4442, New Zealand; 2Complete Feed Solutions, Howick, Auckland 2145, New Zealand

**Keywords:** age, apparent metabolizable energy, soybean meal, canola meal, broilers

## Abstract

**Simple Summary:**

The accurate estimation of the apparent metabolizable energy (AME) content of feed ingredients is crucial for a precise poultry feed formulation. Protein-source ingredients are the second major source of energy supply in broiler diets. The current AME values of protein sources are obtained from reference tables or equations, which have been determined using older broilers (usually 3-5-week-old). In this approach, the effects of broiler age on the AME or nitrogen-corrected AME (AMEn) of feed ingredients are ignored. The objective of the current investigation was to examine the effects of broiler age on the AMEn of soybean meal (SBM) and canola meal (CM) from day 1 to 42 of age. It was found that age has a marked effect on the AMEn of SBM and CM. Nutritionists, therefore, must consider using age-dependent AME or AMEn values for SBM and CM in feed formulation.

**Abstract:**

The effects of broiler age on the apparent metabolizable energy (AME) and nitrogen-corrected AME (AMEn) of soybean meal (SBM) and canola meal (CM) were examined. A corn-SBM basal diet was developed, and two test diets were formulated by substituting (w/w) 300 g/kg of the basal diet with SBM or CM. Six groups of broiler chickens, aged 1–7, 8–14, 15–21, 22–28, 29–35 or 36–42 d post-hatch, were utilized. Each diet, in pellet form, was randomly allocated to six replicate cages in each age group. Except for the 1–7 d age group, the birds were fed a starter (d 1–21) and/or a finisher (d 22–35) diet prior to the introduction of the experimental diets. The number of birds per cage was 10 (d 1–7), 8 (d 8–14) and 6 (d 15–42). The AME and AMEn of SBM and CM were determined by total excreta collection. The data for each protein source were subjected to orthogonal polynomial contrasts using the General Linear Models procedure. Bird age decreased the retention of dry matter quadratically (*p* < 0.001) for both SBM and CM. The retention of nitrogen decreased linearly (*p* < 0.001) with the advancing age of broilers for SBM and CM. The AMEn of SBM and CM decreased quadratically (*p* < 0.001) as the birds grew older. The highest AMEn was observed during week 1 for both SBM and CM, then declined until week 3, followed by increases thereafter. The current results showed that the age of broiler chickens influenced the AMEn of SBM and CM and supported the use of age-dependent AMEn of feed ingredients in diet formulations.

## 1. Introduction

Determining the available energy of feed ingredients for broilers is crucial to optimize their dietary inclusion rate and improving feed efficiency. The most prevalent system in evaluating the energy availability of feed ingredients is the apparent metabolizable energy (AME) or the nitrogen-corrected-AME (AMEn) [1,2]. 

Dietary energy is provided mainly by the starch in cereal grains, accounting for over 60% of the energy requirements for broilers. Plant-based protein sources also contribute up to 30% of the energy supply in broiler diets, and their interaction with main energy sources impacts the overall energy utilization [3,4].

The two plant protein sources commonly used in poultry diets are soybean meal (SBM) and canola meal (CM). Soybean meal remains the sovereign protein source used globally due to its high protein content, excellent amino acid profile that complements cereal grains and high amino acid digestibility [5]. Canola meal contains lower AME than SBM [6] but is increasingly included in broiler diets as an alternative to SBM due to the ever-increasing cost of SBM. 

Commercial nutritionists have been using a single AME or AMEn value of feed ingredients obtained from tabulated values, predictive regression equations or bioassays [7]. This approach ignores the fact that energy utilization is influenced by the number of factors, including the age of birds. Bird age has been reported to influence the digestion and absorption of energy-yielding nutrients in feed ingredients [8,9,10]. During the early growth stages of life, the intestinal tract is immature and less developed [11]. Moreover, low concentrations and low activities of digestive enzymes during the first two weeks of age reduce the ability of birds to digest and utilize nutrients. Studies have shown that the age of birds impacts the AME of complete diets and individual feed ingredients [8,12]. Lopez and Leeson [10] observed that the AMEn of SBM increased by 0.50 MJ/kg between d 12 and d 33 of broiler age. 

The applicability of a single AME value, obtained with older birds, to all growth phases, especially the early life of broilers, could be challenged and underlines the need for age-dependent estimates of AMEn of ingredients in feed formulations. Despite this importance, studies examining the age effects on AMEn of SBM and CM in broilers are scant, and none have investigated the changes from the hatch to the marketing stage. The aim of the present study was to determine the AMEn in SBM and CM from week 1 to 6 post-hatch using the substitution method. This methodology has been used previously in our laboratory to measure the AME and AMEn of four cereal grains [13]. 

## 2. Materials and Methods

The experiment complied with the New Zealand Code of Ethical Conduct for the use of live animals for research and the protocol approved by the University Animal Ethics Committee. 

### 2.1. Ingredients

Two protein sources (SBM and solvent-extracted CM) were obtained from a local commercial supplier. The proximate and nutrient composition of SBM and CM are presented in Table 1. The SBM was of Argentinean origin, and the CM was of Australian origin.

### 2.2. Diets, Birds and Housing 

A total number of 252 one-day-old male broilers (Ross 308) were obtained from a local hatchery and raised on floor pens in an environmentally controlled room until assigned weekly to the experimental treatments. Except for the 1–7 d age group, the birds were fed broiler starter mini pellets (230 g/kg crude protein and 12.56 MJ/kg AMEn) until d 21 and finisher pellets (207 g/kg crude protein and 13.0 MJ/kg AMEn) from d 22 to 35 (Table 2). At the beginning of each week (d 1, 8, 15, 22, 29 and 36), a new batch of birds was selected randomly from the floor pens, weighed individually, and allocated to cages so that the average bird weight per cage was similar. For each protein source, the assay diet was fed to six replicate cages of broilers during the six periods, namely week 1 (d 1–7), week 2 (d 8–14), week 3 (d 15–21), week 4 (d 22–28), week 5 (d 29–35) or week 6 (d 36–42). Each replicate cage housed 10 birds during week 1, 8 birds during week 2, and 6 birds during weeks 3 to 6 post-hatch. 

The AME was determined using the substitution method, as outlined by Khalil et al. [13]. In this method, a corn-SBM basal diet was formulated (Table 2), and then the two test diets were developed by replacing (w/w) 300 g/kg of the basal diet with one of the protein sources. The diets were mixed in a single-screw paddle mixer and then pelleted.

### 2.3. Determination of Metabolizable Energy

The total excreta collection procedure was employed for AME determination [1]. The details of the collection, processing and sampling have been described previously [13]. The diet and excreta samples were analyzed for dry matter (DM), gross energy (GE) and nitrogen (N).

### 2.4. Chemical Analysis

The DM, GE, N, crude fat, ash, neutral detergent fiber and minerals were analyzed, as described previously [13]. The crude protein content was calculated as N × 6.25. 

### 2.5. Calculations

All of the data were expressed on a DM basis, and the AME was determined using the following formula:AME_Diet_ (MJ/kg) = [(FI × GE_Diet_) − (Excreta output × GE_Excreta_)]/Feed intake

The AME of the protein sources was then calculated using the following formula: AME_protein source_ (MJ/kg) = [AME of test protein source diet − (AME of basal diet × 0.70)]/0.30

Nitrogen retention, as a percentage of intake, was determined as follows:N retention (%) = 100 × [((FI × N_Diet_) − (Excreta output × N_Excreta_))/(FI × N_Diet_)]

The AMEn was then calculated by correction for zero N retention by assuming 36.54 KJ per g N retained in the body, as described by Titus et al. [14].

### 2.6. Statistical Analysis

The data for each protein source were analyzed separately by one-way ANOVA using the General Linear Models procedure of the SAS (version 9.4; SAS Institute Inc., Cary, NC, USA). The cages served as the experimental unit. Significant differences between means were separated by the Least Significant Difference test. The data were subjected to orthogonal polynomial contrasts using the General Linear Models procedure of SAS [15] to examine whether the responses to increasing bird age were of linear or quadratic nature. Significance was declared at *p* ≤ 0.05. 

## 3. Results

The influence of broiler age on the retention of DM and N, AME, AMEn and the ratio between AMEn and gross energy of SBM is summarized in Table 3. The retention of DM was the highest in week 1, declined in week 2, plateaued between weeks 3 and 5, and declined further in week 6, resulting in a quadratic (*p* < 0.001) age effect. A linear decrease (*p* < 0.01) in the N retention was observed as birds grew older, from 66.3% in week 1 down to 47.1% in week 6. The AME, AMEn and AMEn:GE of SBM showed a quadratic response with age (Figure 1A). The AMEn of SBM decreased from 11.38 MJ/kg in week 1 to 9.46 MJ/kg in week 3, followed by an increase in week 5 to 10.33 MJ/kg. A similar trend was observed for the AMEn:GE ratio.

The retention of DM and N, AME and AMEn and AMEn:GE of CM measured at different ages of broiler chickens are presented in Table 4. The DM retention of CM showed a quadratic decrease (*p* < 0.001) as birds grew older. Birds retained the highest DM in week 1, decreased to week 2, plateaued between weeks 3 and 5, and declined further in week 6. For N retention, birds retained 66.3% N in week 1, which decreased linearly (*p* < 0.001) to 48.1% in week 6. The AME, AMEn and AMEn:GE of CM decreased quadratically (*p* < 0.001) with advancing age (Figure 1B). The AMEn of CM decreased from 9.10 MJ/kg in week 1 to 6.44 MJ/kg in week 3, increased to 7.30 MJ/kg in week 4, then plateaued up to week 6. 

The influence of broiler age on excreta GE and the ratio between excreta output to FI in birds fed SBM and CM diets are presented in Table 5. There was a quadratic (*p* < 0.001) response to broiler age for excreta GE content and the ratio between excreta output to FI. The GE of excreta increased between week 1 and week 4, followed by a decrease up to week 6 of age. The excreta to FI ratio increased from 0.24 and 0.27 kg:kg at week 1 to 0.36 and 0.39 kg:kg at week 6, respectively, for SBM and CM.

## 4. Discussion

The accurate determination of the metabolizable energy content of feed ingredients is crucial to achieving an optimum energy level based on the energy content of feed ingredients and, therefore, a central factor in least-cost feed formulations. The focus is usually placed on the inclusion level of dietary energy ingredients in the feed formulation, as changes in dietary energy play a pivotal role in determining not only the feed consumption but also the cost of the diet [16,17]. In addition to supplying the majority of dietary protein, protein-source ingredients, such as SBM and CM, are secondary sources of supplying energy in broiler diets after cereals. 

To the authors’ knowledge, no published data are available on age-related energy utilization responses, from week 1 to 6 post-hatch, of individual protein sources. The main objective of the current study was to examine whether the AMEn of SBM and CM is influenced by broiler age and if the age effect varies between these two ingredients. The results showed that the AMEn of SBM and CM decreased with advancing age, with the first week post-hatch recording the highest AMEn value for both protein sources, then declining to the lowest value at d 21 and increasing thereafter to d 42. The retention of DM (quadratically) and N (linearly) also declined with advancing age. Furthermore, the ratio of AMEn:GE declined between d 7 and d 21 and increased with the advancing age of the broilers. 

The observed reductions in AMEn and the reduced DM and N with the advancing age of the broilers were unexpected as it is well accepted that the digestive tract of newly hatched chicks is immature and lacks the appropriate enzymatic capabilities for efficient nutrient digestion and absorption. The rapid development and maturation of the digestive system, the increased secretion of digestive enzymes and morphological surface area for nutrient absorption, along with improvements in nutrient transporter activities, are observed as the broilers grew older [11,18].

The influence of broiler age on the AMEn of SBM and CM has not been previously evaluated. The published data on the influence of age on the AME of complete diets, although mostly limited to two age groups, present contradictory patterns. Some studies showed an increasing trend [8,10,19,20], while others revealed a decrease [21,22]. In some studies, no influence of age was observed [23,24,25,26]. Batal and Parsons [27] showed that the AMEn of a corn-SBM diet increased from 13.33 MJ/kg at d 7 to 14.35 MJ/kg at d 14, with no further increase up to d 21 post-hatch. Yang et al. [28] found that there was a linear AMEn response to age, with AMEn increasing from d 7 to 28 and then slightly decreasing at 35 d of age. Kras et al. [29] found that the AMEn of diets with low and high fiber contents decreased by 0.24–0.30 MJ/kg, respectively, between d 10 and 20 post-hatch. The higher AMEn values at d 10 compared to d 20 were attributed to the longer digesta retention time and, therefore, the better digestion of nutrients. Bartov [12] reported a decline in dietary AMEn of 0.32 MJ/kg with broiler age between d 13 and 22 post-hatch. The reduction was ascribed mainly to the diet composition, specifically to the dietary ratio between crude protein and energy. 

In agreement with the present findings, Thomas et al. [30] reported that the AMEn of a wheat-based diet decreased from 13.90 MJ/kg at d 3 to 12.17 MJ/kg at d 5 post-hatch, with a further decrease to 11.06 MJ/kg at d 7. Similarly, it was observed that the AMEn of a sorghum-based diet declined between d 3 and d 5 (14.49 vs. 12.76 MJ/kg), followed by a 0.62 MJ/kg reduction at d 7 post-hatch. Moreover, the AMEn of a corn-based diet was reduced with advancing age from 13.87 MJ/kg at d 3 to 12.28 at d 7 of age. These researchers also reported an increase in dietary AMEn from d 7 to 14 post-hatch. Regardless of the cereal base, the highest N retention and total tract fat digestibility were recorded at d 3, which decreased gradually to the lowest values at d 7. The exact cause for this reduction in the utilization of energy-yielding components towards the end of the first week is unclear. However, as discussed below, a number of factors may have contributed to this finding, including the yolk sac contribution to dietary energy, changes in the microbiota, the availability of digestive enzymes, digesta passage rate and endogenous energy losses (EEL). 

Zelenka [31] revealed that the AME of a corn-SBM diet was the highest at d 3 post-hatch, declined to the lowest value by d 7, followed by an increase at d 14 post-hatch. In a follow-up study, this researcher reported a similar reduction in dietary AME from hatch to 7 d of age, followed by an increase at 14 d post-hatch. Murakami et al. [32] reported a decline in energy utilization during the first few d post-hatch, followed by an increase after d 7. Gracia et al. [33] reported that the dietary AME declined between d 4 and 8, which was associated with a reduction in the retention of N and fats on d 8, followed by an increase in dietary AMEn on d 15 post-hatch. 

In the current study, the ratio between AME and GE for both SBM and CM declined with the advancing age of the broilers. A similar trend was reported by Moss et al. [34], who showed that the AMEn:GE ratio for a corn-based diet was higher at d 7 than at d 34 (0.807 vs. 0.778). Similar trends were observed for sorghum-based (0.778 vs. 0.718) and wheat-based (0.798 vs. 0.761) diets.

The higher AMEn of SBM and CM at d 7 post-hatch may be attributed to a combination of three factors: First, FI is very low during the first week compared to subsequent weeks. Lower FI restricts the uniform flow of digesta in the digestive tract and increases the digesta retention time, allowing better digestion and absorption of nutrients [35,36]. With the advancing age of the broilers, the rapid increase in FI could lead to shorter digesta retention time in the digestive tract, which decreases the duration of contact with digestive enzymes, hence reducing nutrient digestibility and utilization [30]. Vergara et al. [37] related the increase in the digesta passage rate to the increase in FI as the birds grew older. Uni et al. [38] observed that the digesta passage time increased from 74 min to 122 min between d 7 and 14 of age, respectively. In agreement, Rougiere and Carre [39] demonstrated that the digesta passage rate increased by 25% between d 9 and 29 of age for broiler chickens. Moreover, at low levels of FI, the EEL (measured as g/kg FI) will be proportionally higher than at higher FI levels, which increases the AMEn of feed ingredients [40]. Murakami et al. [41] similarly stated that the EEL estimates increased with decreasing FI levels.

The gastrointestinal tract (GIT) matures as birds grow older and the secretion and activities of enzymes increase, improving the digestibility of nutrients [42,43]. However, the development of GIT and increases in the digestive and enzymatic activities might not be able to keep up with the marked increase in FI with the advancing age of the broilers, reducing energy utilization compared to the first week. Moreover, the relative contribution of FI to the excreted energy content at low FI is higher than at higher FI [44]. As observed in the current study, the FI to excreta output ratio for SBM and CM increased quadratically with the advancing age of broilers. The lowest ratio of excreta to FI for both SBM and CM diets was recorded on d 7, suggesting that fewer nutrients are excreted per unit of the feed consumed during the first week, which might have contributed to the higher AMEn on d 7. 

Second, the presence of the residual yolk sac could have contributed to the higher AMEn during week 1. The lipids of the yolk provide the chick embryo with over 90% of its energy requirement for development [45]. Yolk lipids are utilized extensively during the last week of incubation, but 25% still remain unutilized at the time of hatch [45]. The residual yolk sac plays an important role in the overall nutrition, growth and development during the first few days post-hatch. Chamblee et al. [46] showed that body weight significantly increased only after the absorption of 20% of the residual yolk sac. Ravindran and Abdollahi [18] discussed that the presence of the residual yolk sac might be beneficial for the utilization of protein and energy. It has been speculated that the residual yolk sac contributes to the breakdown of lipids through the lipolytic enzymes, providing 90% of the total energy required for the hatching and 30% of the energy required during the first 3 d post-hatch [47,48,49]. This enzymatic influence of the yolk sac towards lipid digestion could have extended for a period after hatch. The absorption through the membrane of the yolk sac via direct release into the blood circulation and/or the expulsion through the yolk stalk into the GIT are some possible explanations for the beneficial effects of the yolk sac on lipid utilization. However, the exact mechanism or contribution of the yolk sac towards nutrient utilization remains unclear [18,50]. 

The third possible explanation for the reductions in AMEn of SBM and CM with the advancing age of the broilers could be related to the development of the microbiota. During the first week post-hatch, the sterile intestinal environment and the absence of microbial population in the neonatal chick may, in part, provide apparent advantages in terms of nutrient utilization and AME. As the birds grow older, the intestinal microbiome population increases, hence competing for energy and nutrients from diets resulting in a reduction in the AMEn of diets [30]. Substitution is the commonly used method for the measurement of AME of protein sources. The direct method is unsuitable because of the issues of palatability and anti-nutritional factors. The regression method could be used but it is costly. The published data on the AME of SBM and CM have generally used assay diets wherein the ingredient was substituted at 200–300 g/kg and with older broilers (21–35 d of age). In the current study, the AME of SBM during 21–35 d ranged from 10.17–11.64 MJ/kg, and those of CM ranged from 7.27–9.22 MJ/kg. The AME values for SBM are higher, and those for CM compare closely with those reported in the literature. Ravindran et al. [51] showed that the AME of SBM from different origins ranged from 8.39–9.94 MJ/kg. Olukosi [52] revealed that the AME of SBM was 10.07 MJ/kg at 21 d of age. Ahiwe et al. [53] discussed that the AME of CM ranged from 8.39–9.15 MJ/kg. Khajali and Slominski [6] showed that the AME of CM varied among different canola varieties ranging from 7.27 to 9.16 MJ/kg. The lower AME content of CM, compared to SBM, is likely due to differences in oligosaccharides (2.0 vs. 5.6%, respectively) and fiber contents (11.2 vs. 5.4%, respectively). 

The current findings showed that age influences the AMEn partly through its effect on the digestion and utilization of energy-yielding nutrients, wherein the retention of DM and N is reduced with the advancing age of the broilers. Bartov [12] showed that the retention of DM decreased by 2.4% between d 13 and 22 of broiler age. It was also shown that the N retention decreased from 60.1% at d 13 to 57.9% at d 22. Yang et al. [28] reported that DM retention increased with age from 75.3% at d 7 to 78.4% at d 28, then declined to 76.2% at d 35 of age. These researchers also reported that the N retention decreased only between d 28 and d 35 from 82.5% to 81.8%, respectively. Fonolla et al. [26] demonstrated that the N retention was higher for younger birds (21–26 d) than in older birds (52–57 d) in a corn SBM diet (60.7% vs. 54.0%). Moss et al. [34] observed a reduction in the retention of N between young (7–9 d) and older broilers (33–34 d) in corn-based (70.7 vs. 61.7%), sorghum-based (70.4 vs. 57.3%) and wheat-based (69.4 vs. 60.2%) diets.

## 5. Conclusions

The present findings demonstrate that the AME and AMEn of SBM and CM were influenced by the age of the broilers. The AME was determined to be highest during week 1 compared to subsequent weeks. This finding was contrary to expectations and could be ascribed to low FI, longer digesta retention time, yolk sac contribution to dietary energy and changes in the numbers and the diversity of the microbiome. Among these possibilities, it is tempting to speculate that the FI intake is the major contributing factor, but further research is warranted to establish and better understand this effect. These findings confirm that the use of a single AME value of feed ingredients in diet formulations is questionable, and age-dependent AMEn values need to be considered in formulations to optimize economic returns. 

## Figures and Tables

**Figure 1 animals-13-00219-f001:**
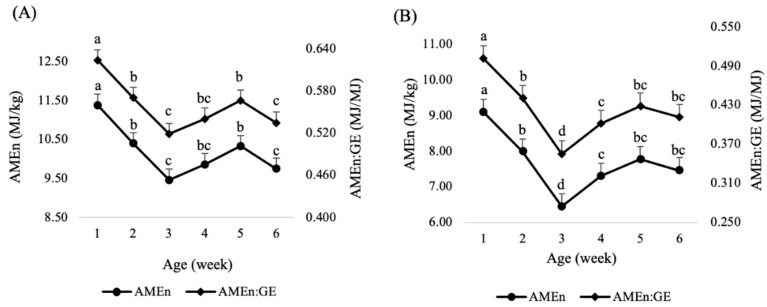
Effect of broiler age on the nitrogen-corrected apparent metabolizable energy (AMEn) and the ratio between AMEn and gross energy (AMEn:GE) for soybean meal (**A**) and canola meal (**B**); mean ± standard error. a–d Values with different superscripts differ significantly (*p* < 0.05).

**Table 1 animals-13-00219-t001:** Proximate, carbohydrate and mineral composition of the tested protein sources (g/kg; as received basis).

Item	Soybean Meal	Canola Meal
DM ^a^	921	911
Ash	65.4	72.5
Nitrogen	77.4	56.9
Protein	489	356
Fat	12.3	48.5
NDF ^a^	84.6	250
Ca ^a^	3.26	5.78
P ^a^	6.61	10.5
GE ^a^ (MJ/kg)	17.67	17.83

^a^ Ca, calcium; DM, dry matter; GE, gross energy; NDF, neutral detergent fiber; P, phosphorus.

**Table 2 animals-13-00219-t002:** Composition (g/kg, as fed basis) of the basal diet used in the apparent metabolizable energy assay and, of pre-assay diets fed to broiler starters (d 1 to 21) and finishers (d 22 to 35) ^1^.

Ingredient	Basal Diet	Starter Diet	Finisher Diet
Corn	604.4	574.2	660.0
Soybean meal, 460 g/kg	338.1	381.4	295.7
Soybean oil	14.2	8.8	13.6
Dicalcium phosphate	15.8	10.7	8.2
Limestone	10.4	11.3	9.9
L Lysine HCl	3.7	2.0	1.9
DL Methionine	3.1	3.3	3.0
L Threonine	2.0	1.0	0.7
L Valine	0.7	-	-
Sodium chloride	1.0	2.5	2.5
Sodium bicarbonate	3.9	2.7	2.5
Trace mineral premix ^2^	1.0	1.0	1.0
Vitamin premix ^2^	1.0	1.0	1.0
Choline Chloride 60%	0.7	-	-
Ronozyme HiPhos (Phytase)	-	0.1	0.1

^1^ The same basal, starter and finisher diets were used in a previous AME assay [13]. ^2^ Vitamin and trace mineral premix supplied the following per kilogram of diet: antioxidant, 100 mg; biotin, 0.2 mg; calcium pantothenate, 12.8 mg; vitamin D3 (cholecalciferol), 2400 IU; cyanocobalamin, 0.017 mg; folic acid, 5.2 mg; menadione, 4 mg; niacin, 35 mg; pyridoxine, 10 mg; vitamin A (trans-retinol), 11100 IU; riboflavin, 12 mg; thiamine, 3.0 mg; vitamin E (dl-α-tocopheryl acetate), 60 IU; choline chloride, 638 mg; Co, 0.3 mg; Cu, 3.0 mg; Fe, 25 mg; I, 1 mg; Mn, 125 mg; Mo, 0.5 mg; Se, 200 µg; Zn, 60 mg.

**Table 3 animals-13-00219-t003:** Influence of broiler age on the retention (% of intake) of dry matter (DM) and nitrogen (N), apparent metabolizable energy (AME; MJ/kg DM), nitrogen-corrected AME (AMEn; MJ/kg DM) and the ratio between AMEn and gross energy (AMEn:GE; MJ/MJ) of soybean meal ^1^.

Age (Week)	DM Retention	N Retention	AME	AMEn	AMEn:GE
1	76.2	66.3	12.99	11.38	0.624
2	70.9	58.3	11.70	10.40	0.570
3	66.8	55.9	10.17	9.46	0.519
4	67.3	56.6	10.98	9.86	0.541
5	68.0	56.3	11.64	10.33	0.567
6	63.9	47.1	10.60	9.75	0.535
SEM ^2^	0.55	1.10	0.280	0.191	0.0105
Orthogonal polynomial contrast, *p* ≤					
Linear	0.001	0.001	0.001	0.001	0.001
Quadratic	0.001	0.811	0.001	0.001	0.001

^1^ Each value represents the mean of six replicates. The number of birds per replicate cage was 10 (week 1), 8 (week 2) and 6 (weeks 3–6). ^2^ Pooled standard error of mean.

**Table 4 animals-13-00219-t004:** Influence of broiler age on the retention (% of intake) of dry matter (DM) and nitrogen (N), apparent metabolizable energy (AME; MJ/kg DM), nitrogen-corrected AME (AMEn; MJ/kg DM) and the ratio between AMEn and gross energy (AMEn:GE; MJ/MJ) of canola meal ^1^.

Age (Week)	DM Retention	N Retention	AME	AMEn	AMEn:GE
1	72.9	66.3	10.56	9.10	0.501
2	68.7	63.2	9.53	7.99	0.440
3	63.0	56.9	7.27	6.44	0.355
4	63.9	58.5	8.30	7.30	0.402
5	64.8	58.7	9.22	7.78	0.428
6	61.0	48.1	8.18	7.46	0.411
SEM ^2^	0.58	0.81	0.348	0.229	0.0127
Orthogonal polynomial contrast, *p* ≤					
Linear	0.001	0.001	0.001	0.001	0.001
Quadratic	0.001	0.140	0.001	0.001	0.001

^1^ Each value represents the mean of six replicates. The number of birds per replicate cage was 10 (week 1), 8 (week 2) and 6 (weeks 3–6). ^2^ Pooled standard error of mean.

**Table 5 animals-13-00219-t005:** Influence of broiler age on excreta gross energy (GE; MJ/kg DM) and excreta output:feed intake (kg:kg); in birds fed soybean meal and canola meal diets ^1^.

Age (Week)	Soybean Meal		Canola Meal	
	Excreta GE (MJ/kg)	Excreta Output: Feed Intake	Excreta GE (MJ/kg)	Excreta Output: Feed Intake
1	15.79	0.24	16.29	0.27
2	16.12	0.29	16.80	0.31
3	16.12	0.33	16.62	0.37
4	15.80	0.33	16.33	0.36
5	15.56	0.32	15.99	0.35
6	15.44	0.36	15.94	0.39
SEM ^2^	0.048	0.006	0.079	0.006
Orthogonal polynomial contrast, *p* ≤				
Linear	0.001	0.001	0.001	0.001
Quadratic	0.001	0.001	0.001	0.001

^1^ Each value represents the mean of six replicates. The number of birds per replicate cage was 10 (week 1), 8 (week 2) and 6 (weeks 3–6). ^2^ Pooled standard error of mean.

## Data Availability

All of the available data are incorporated in the manuscript.

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
