# Peer review of "Broiler Age Influences the Apparent Metabolizable Energy of Soybean Meal and Canola Meal"

_animals, 2023, doi:10.3390/ani13020219_

Round 1

Reviewer 1 Report

It was a pleasure reviewing your article.

Author Response

Thanks for your nice comments

Reviewer 2 Report

This manuscript described the effect of age on the AME of SBM and CM.  It provides a new information for the researchers in the field of poultry nutrient.  Their findings proofed that the AME of protein source can be related to different stage of growth or age.  There are a few suggestions:

Table one provided the composition of the tested protein source and table 2 provided the composition of the diets.  However, the composition of the tested diets was not provided.  It is suggested that the composition of the tested diets should be calculated or analyzed and listed.  

In table 5 and line the description in line 191-192, authors described a linear age effect on the excreta to FI ratio, but the quadratic effect were both 0.001.  Please explain why the data looked like a linear response, but the statistic outcome showed a significant quadratic effect.

Author Response

Table one provided the composition of the tested protein source and table 2 provided the composition of the diets.  However, the composition of the tested diets was not provided.  It is suggested that the composition of the tested diets should be calculated or analyzed and listed.  

Response: Thanks for the comment. The test diets was formulated following the substitution method, where 30% of the basal diet was replaced by the test protein source. Since composition of both the test ingredient and basal diet are known, the composition of test diets could arithmetically calculated.

In table 5 and line the description in line 191-192, authors described a linear age effect on the excreta to FI ratio, but the quadratic effect were both 0.001.  Please explain why the data looked like a linear response, but the statistic outcome showed a significant quadratic effect.

Response: The magnitude of changes between different ages vary, causing the quadratic effect. Thus this effect takes precedence over the linear effect

Reviewer 3 Report

That’s a very nice and well-written paper, with a clearly stated problem and objective, correctly prepared results, good discussion and conclusion.

The topic is interesting and of a practical nature for poultry nutrition and thus the paper is worth of publication. I have a few questions for clarification, some technical comments and some suggestions.

Abstract

L25-26: To me it is not clear when you say: “birds were fed a starter (d1-21) and/or finisher (d22-35) diet prior to the introduction of experimental diets”; but then if birds were fed from d 1-21 a starter diet, when did those in group 8-14 d receive experimental diet? This can be re-written in a clearer way.

Diets, birds and housing

L87: Any information about corn particle size or grinding before pelleting? Was corn finely ground? 2mm? 4mm sieve?

Determination of metabolizable energy

L115: was there any contamination when excreta was collected? Wood shavings? Feathers etc..

Figure 1

Just a suggestion: for the sake of comparison, maybe have the same unit on the x axis for both protein sources

Discussion

L244: AMEn of corn-based diet was? reduced with advancing age…

L268: What does “Lower FI restricts the uniform flow of digesta in the digestive tract” mean?

E.g. if feed was finely ground and fed as pellets, how will the flow of fine particles through the digestive tract be restricted? Is feed flow in the digestive tract always uniform?

- In both references 36 and 37, the findings are not in line with the mentioned statement on L 268-269. Please double check this.

- In reference 36: the finding reported was: "no relationship of restriction of feed or amount of feed consumption to feed passage rate."

- In reference 37, they did not study feed restriction or assessed different feed intake level on digesta retention time. Please double check this.

L270-272: Here you state that as birds age, their feed intake increases, and so does digesta passage rate (i.e. feed passes more quickly throughout the GIT), which results in a reduction in digestibility (due to shorter contact duration with enzymes). In Uni et al's study, passage time was longer at older age (74 min to 122min), so according to your previous sentence (With advancing age....) this is the opposite.

L273-275 in this study, passage time and not passage rate was assessed, and not to 22 days but until 14 days (4, 7, 10 and 14 d). Please double check this.

L276-278 this statement contradicts what is mentioned in L268 (with a low level of FI, there is a decrease in AMEn vs lower FI allows better digestion and absorption). What does this example support exactly? that AMEn at younger age is higher or lower compared to old age?

L288 was recorded on or at d 7

L319 How do the findings compare closely with values in the literature for SBM: The range between the reported and literature is a bit wide. Current SBM AME values 10.17-11.64 vs literature example 8.39-9.94 MJ/kg. Please double check this.

Can you speculate (after discussing the potential factors for the reduced AME after week 1) which of the three factors contribute the most to the reduction in AME? Or do the three factors contribute equally to this observation?

Author Response

L25-26: To me it is not clear when you say: “birds were fed a starter (d1-21) and/or finisher (d22-35) diet prior to the introduction of experimental diets”; but then if birds were fed from d 1-21 a starter diet, when did those in group 8-14 d receive experimental diet? This can be re-written in a clearer way. 

Response: Thanks for your comment. The starter or the finisher diet was provided for the spare birds which were not selected for the experimental diets weekly. Only the group of birds selected every week were fed the experimental diets.

L87: Any information about corn particle size or grinding before pelleting? Was corn finely ground? 2mm? 4mm sieve?

Response: The corn was of 3.0mm particle size

L115: was there any contamination when excreta was collected? Wood shavings? Feathers etc.

Response: Thanks for your comment. Birds were housed in cages; therefore no contamination from wood shaving. Other sources of contaminations were minimized as possible by removing feathers and feed dropped on the trays.

 - Just a suggestion: for the sake of comparison, maybe have the same unit on the x axis for both protein sources

Response: Same units are used for both protein sources. The x axis in both figures represents the age (week)

L244: AMEn of corn-based diet was reduced with advancing age…

Response:  Revised as suggested.

L268: What does “Lower FI restricts the uniform flow of digesta in the digestive tract” mean? E.g. if feed was finely ground and fed as pellets, how will the flow of fine particles through the digestive tract be restricted? Is feed flow in the digestive tract always uniform?

- In both references 36 and 37, the findings are not in line with the mentioned statement on L 268-269. Please double check this. - In reference 36: the finding reported was: "no relationship of restriction of feed or amount of feed consumption to feed passage rate." .- In reference 37, they did not study feed restriction or assessed different feed intake level on digesta retention time. Please double check this.

Response: Thanks for this comment. With lower FI, the digesta flow will be slower which increases the retention time of digesta in the digestive tract, resulting in increases in nutrient digestibility and AME. In one of our previous publications (Khalil et al., 2021), feed form was shown to impact theFI and AME of protein sources. The uniformity of feed flow depends on several factors including feed intake.

Both references 36 and 37 have been deleted and updated with new references.

L270-272: Here you state that as birds age, their feed intake increases, and so does digesta passage rate (i.e. feed passes more quickly throughout the GIT), which results in a reduction in digestibility (due to shorter contact duration with enzymes). In Uni et al's study, passage time was longer at older age (74 min to 122min), so according to your previous sentence (With advancing age....) this is the opposite.

Response:  In Uni et al’s study, the passage rate was 74 mins at 7 d of age and then increased to 122 mins at 14 d of age, which indicates that the passage rate increases with age.

L273-275  n this study, passage time and not passage rate was assessed, and not to 22 days but until 14 days (4, 7, 10 and 14 d). Please double check this.

Response:  Thanks for the comment. Sentence has been corrected.

 L276-278 this statement contradicts what is mentioned in L268 (with a low level of FI, there is a decrease in AMEn vs lower FI allows better digestion and absorption). What does this example support exactly? that AMEn at younger age is higher or lower compared to old age?

Response:  Statement in L268, explained that with lower FI, digesta passage rate decreases, which increase the digestion and absorption of nutrients, hence increase AMEn. The statement in L276-278 highlights that increasing EEL attributed to lower FI has influence AMEn value of feed ingredients.

L288 was recorded on or at d 7

Response:  Corrected

L319 How do the findings compare closely with values in the literature for SBM: The range between the reported and literature is a bit wide. Current SBM AME values 10.17-11.64 vs literature example 8.39-9.94 MJ/kg. Please double check this.

Response:  Agree. Sentence revised.

Can you speculate (after discussing the potential factors for the reduced AME after week 1) which of the three factors contribute the most to the reduction in AME? Or do the three factors contribute equally to this observation? 

Response: FI probably contributed more to the reduction AME of protein sources. However, this statement needs more research to be verified. Additional statements included now.